# Persistence as a Robust Indicator of Medication Adherence-Related Quality and Performance

**DOI:** 10.3390/ijerph18094872

**Published:** 2021-05-03

**Authors:** Enrica Menditto, Caitriona Cahir, Sara Malo, Isabel Aguilar-Palacio, Marta Almada, Elisio Costa, Anna Giardini, María Gil Peinado, Mireia Massot Mesquida, Sara Mucherino, Valentina Orlando, Carlos Luis Parra-Calderón, Enrique Pepiol Salom, Przemyslaw Kardas, Bernard Vrijens

**Affiliations:** 1CIRFF, Center of Pharmacoeconomics and Drug Utilization Research, Department of Pharmacy, University of Naples Federico II, 80131 Naples, Italy; sara.mucherino@unina.it (S.M.); valentina.orlando@unina.it (V.O.); 2Data Science Centre, Royal College of Surgeons in Ireland, D02 YN77 Dublin, Ireland; caitrionacahir@rcsi.ie; 3Preventive Medicine and Public Health Department, Zaragoza University, Fundación Instituto de Investigación Sanitaria de Aragón (IIS Aragón), 50009 Zaragoza, Spain; smalo@unizar.es (S.M.); iaguilar@unizar.es (I.A.-P.); 4UCIBIO/REQUIMTE, Competences Centre on Active and Healthy Ageing of the University of Porto, Porto4Ageing, Faculty of Pharmacy, University of Porto, 4099-002 Porto, Portugal; marta.almada@reit.up.pt (M.A.); emcosta@ff.up.pt (E.C.); 5IT Department, Istituti Clinici Scientifici Maugeri IRCCS Pavia, Pavia 27100, Italy; anna.giardini@icsmaugeri.it; 6Drug Information Centre and Pharmaceutical Care Department, Muy Ilustre Colegio Oficial de Farmacéuticos de Valencia (MICOF Valencia), 46003 Valencia, Spain; maria.gil@micof.es; 7Servei d’Atenció Primària Vallès Occidental, Institut Català de la Salut, 08202 Barcelona, Spain; mmassot.mn.ics@gencat.cat; 8Group of Research and Innovation in Biomedical Informatics, Biomedical Engineering and Health Economy, Institute of Biomedicine of Seville, IBiS/Virgen del Rocío University Hospital/CSIC/University of Seville, 41004 Sevilla, Spain; carlos.parra.sspa@juntadeandalucia.es; 9International Committee, Muy Ilustre Colegio Oficial de Farmacéuticos de Valencia (MICOF Valencia), 46003 Valencia, Spain; e.pepiol.000@micof.es; 10Medication Adherence Research Centre, Medical University of Lodz, 90-136 Lodz, Poland; przemyslaw.kardas@umed.lodz.pl; 11AARDEX Group & Liège University, 4000 Liège, Belgium; bernard.vrijens@aardexgroup.com

**Keywords:** medication adherence, persistence, quality of care, performance indicator, electronic health records

## Abstract

Medication adherence is a priority for health systems worldwide and is widely recognised as a key component of quality of care for disease management. Adherence-related indicators were rarely explicitly included in national health policy agendas. One barrier is the lack of standardised adherence terminology and of routine measures of adherence in clinical practice. This paper discusses the possibility of developing adherence-related performance indicators highlighting the value of measuring persistence as a robust indicator of quality of care. To standardise adherence and persistence-related terminology allowing for benchmarking of adherence strategies, the European Ascertaining Barriers for Compliance (ABC) project proposed a Taxonomy of Adherence in 2012 consisting of three components: initiation, implementation, discontinuation. Persistence, which immediately precedes discontinuation, is a key element of taxonomy, which could capture adherence chronology allowing the examination of patterns of medication-taking behaviour. Advances in eHealth and Information Communication Technology (ICT) could play a major role in providing necessary structures to develop persistence indicators. We propose measuring persistence as an informative and pragmatic measure of medication-taking behaviour. Our view is to develop quality and performance indicators of persistence, which requires investing in ICT solutions enabling healthcare providers to review complete information on patients’ medication-taking patterns, as well as clinical and health outcomes.

## 1. Introduction

It is widely recognized that medication adherence is a key component of quality of care for disease management. Thus, improving adherence with treatment is a priority for health systems worldwide [1,2,3,4]. According to a 2003 report published by the WHO [5], in developed countries, on average, only 50% of patients are adherent with their prescribed medications. Medication adherence is a key factor associated with the effectiveness of all pharmacological therapies but is particularly critical for medications used for chronic conditions. Several studies have shown that lower levels of adherence and more specifically poor persistence with treatment are associated with higher healthcare costs, poorer health outcomes and lower patient quality of life and satisfaction, as well as increased disease prevalence and relapse [6,7]. A recent report of the OECD [8] investigating health systems efficiency stated that ‘*routine medication adherence measures as well as adherence-related quality and performance indicators should be encouraged in order to improve health system effectiveness and efficiency*’.

Measuring the quality of care in disease management has become an increasingly important part of evaluating and improving healthcare delivery [8]. Measuring and reporting performance indicators allows making policy priorities explicit, defining responsibilities/expectations, facilitating accountability, and focusing resources [8]. Therefore, performance indicators are measures that capture a variety of health- and health system-related trends and factors based on an operational definition of quality [9]. They can be difficult to operationalise because essentially, they are quantitative measures of quality and quality is a multidimensional construct, based on numerous and sometimes conflicting approaches [10]. One of the best-known approaches is the Donabedian three-part model [11], where healthcare quality is assessed based on the structure (resources of the healthcare system), process (what healthcare providers/patients do) and outcome (health, economic) of the healthcare system. Each part of the model is interdependent, with good structures promoting good processes and in turn good processes promoting good outcomes.

To date, adherence-related quality and performance indicators have been rarely explicitly included in national health policy agendas. According to the OECD report, very few countries routinely measure and report adherence as a quality improvement indicator or performance measures at the system level. [8] The United States and Sweden are the only OECD countries that measure and report on adherence and persistence on a routine basis at the health system level and only for cardiovascular disease [8]. One of the possible reasons for this is the lack of standardised adherence terminology and use of routine measures of adherence in clinical practice [6]. This has also limited the use of Big Data in developing monitoring systems capable of reporting timely, reproducible and accurate information on medication-taking behaviour. This paper discusses the possibility of developing adherence-related quality and performance indicators [8]. In particular, the value of measuring persistence with treatment as a robust and sound indicator of quality of care within healthcare systems is highlighted.

## 2. Development of Adherence-Related Performance Indicators

In order to develop adherence-related performance indicators to improve disease management, we need to determine (i) what we are measuring, (ii) how can it be measured and (iii) the scientific robustness (reliability and validity) of the measure [12].

### 2.1. What Are We Measuring?

In an effort to define and standardise adherence and persistence-related terminology for clinical and research use and to allow for benchmarking of existing adherence enhancing strategies, the European Ascertaining Barriers for Compliance (ABC) project proposed a new Taxonomy of Adherence [13]. The Taxonomy defines adherence as the process by which patients take their prescribed medications and consists of three essential components: (i) initiation; (ii) implementation; (iii) discontinuation. The process starts with initiation when the patient takes the first dose of a prescribed medication. The process continues with the implementation of the dosing regimen, defined as the extent to which a patient’s actual dosing corresponds to the prescribed dosing regimen, from initiation until the last dose is taken. Discontinuation marks the end of therapy, when the next dose to be taken is omitted and no more doses are taken thereafter. A key element of the Taxonomy defines *persistence* as the length of time between initiation and the last dose, which immediately precedes discontinuation (Figure 1). After discontinuation, there may be a period of non-persistence until the end of the prescribing period [14].

### 2.2. How Can It Be Measured?

Administrative databases such as pharmacy claims data, patients’ health records and laboratory files provide a non-invasive, objective and relatively inexpensive method to estimate adherence at the population level in real-world settings [15,16,17]. Administrative databases can be linked through pseudo-anonymized patient codes to establish, for example, the association between adherence and clinical and health outcomes [18]. However, there is no standardised method of measuring the three different components of the adherence taxonomy (i.e., initiation, implementation and discontinuation) using pharmacy claims data. In order to measure *initiation*, the prescribing and dispensing events need to be assessed together, but there is often a lack of necessary data linkage between what is prescribed by the doctor and dispensed by the pharmacist on a large scale, outside of a small number of integrated healthcare systems [19,20,21]. Moreover, a well-documented act of dispensation is not sure to lead to true initiation of therapy, as some patients may not take the first dose once dispensed.

The implementation component of adherence is often estimated by calculating a type of summary statistic. Metrics such as proportion of days covered (PDC) and medication possession ratio (MPR) are frequently used to summarise overall adherence as the percentage of a treatment regimen that a patient has likely taken as prescribed (e.g., 50% or 80%) based on the number of days that medication is dispensed for during a specified time period [22]. *Implementation* is often classified dichotomously, with ratios above a specified threshold denoting adherence [23]. However, there is no consensual standard for what constitutes adequate *implementation*. Many studies consider 80% to be acceptable, whereas 95% is considered mandatory when the treatment is unforgiving for minor deviations in medication adherence [24]. Overall quality indicators, such as the MPR and the PDC, also provide a wrong estimation in situations such as drug oversupply or stockpiling by patients [25]. The main problem with the estimation of PDC and MPR from administrative databases comes from its sparse frequency of sampling, typically every 3 months. Therefore, these measures only provide an aggregate summary of treatment availability regardless of treatment *discontinuation*.

### 2.3. Scientific Robustness of the Measure

A reliable adherence performance indicator should provide a consistent measure of adherence in similar populations or settings. Reported adherence rates are known to vary widely. *Non-initiation* has been shown to vary between 2.3 and 50% across studies (weighed average = 5.1 ± 1.3%) [19,20,21], while *implementation* has been shown to vary between 4 and 92%, with the generally accepted understanding that 50% of treatments for chronic conditions are not taken as prescribed [5]. There is a great variety in the literature regarding the definition for the appropriate length of the permissible gap in *discontinuation* and it has been reported to range between 15 and 120 days after the end of the previous refill [13]. This large variation in adherence rates may reflect the different methods of measurement and time frames applied in studies.

An adherence performance indicator should measure what it is supposed to measure and have a casual association with clinical outcomes or healthcare resource use through scientific evidence, in order to be valid. The selection of the cut-off point for *implementation* adherence should require that taking, e.g., ≥80% of the medication leads to better clinical outcomes than taking less than 80%. However, a recent systematic review [26] was unable to confirm or reject the validity of the commonly used 80% threshold. 

## 3. Measuring Persistence: A More Useful Indicator?

A more robust, informative, and feasible way to measure adherence using pharmacy claims data could be to measure *persistence* with treatment. *Persistence* represents the time (e.g., days, months, years) over which a patient continues the treatment. For practical reasons, it might be assessed according to the time taken for a patient to fill their prescription and can capture both the timeliness and frequency of refilling [13]. In reality, as defined by the adherence taxonomy, adherence is a dynamic behaviour, consisting of *initiation*, *implementation* and *discontinuation* phases of treatment that vary over time, resulting in periods of persistence and non-persistence [13,26]. Therefore, rather than measuring the specific components of adherence, we could measure *persistence*, which captures the chronology of adherence and enables us to examine and understand patterns of medication-taking behaviour [27].

Indeed, group-based trajectory modelling (GBTM) has been increasingly used in adherence research [28,29,30]. This methodology has shown that an average value of adherence (e.g., PDC < 50%) can be assigned to participants who have very different patterns of medication-taking behaviour during a short period of time, including those who consistently have treatment gaps and those with initial poor adherence that improves over time [28,29,30]. A recent systematic review of medication adherence trajectories identified 4 to 6 trajectory groups that described different longitudinal medication adherence behaviours [31]. In this scenario, *persistence* may be a more appropriate and feasible indicator of the quality of disease management. However, similar to other existing measures of adherence (MPR, PDC) using pharmacy claims data, persistence is not free from certain limitations. In particular, this measure ignores whether a patient actually administers/takes the medication as prescribed or not. A US study on type 2 diabetes (T2D) assessed adherence through both PDC and persistence, allowing a 45-day gap between two prescriptions, and highlighted that the pharmacy claims used indicated only that a prescription was filled and it remained unknown whether patients used the medication as prescribed [32].

## 4. Real-World Challenges

If measures of medication persistence are to be used as performance indicators to monitor, benchmark and evaluate the quality of disease management, there are a number of challenges which need to be addressed. The use of routine data provides an efficient way to monitor persistence but there are methodological challenges in using pharmacy claims data and electronic medical records as an information source for monitoring persistence. Advances in eHealth and Information Communication Technology (ICT) could play a large role in providing the necessary structures to develop indicators of persistence [33]. Data used to develop indicators of persistence should be standardised, transparently communicated and shared between similar clinical areas, with uniform definitions that support the measurement process and facilitate meaningful comparison. This can be achieved through the development of a minimum data set (MDS) containing a list of standardised data which can provide a uniform approach to conducting a comprehensive assessment.

The indicators must also appropriately reflect and account for the variations in medication persistence within the context of the healthcare setting. Medication persistence is influenced by a number of interconnected factors related to the patient, the provider, and the healthcare system [5]. Moreover, there are several factors beyond the control of the patient, provider and health system that influence persistence, e.g., socioeconomic status, education, the environment as well as the costs of therapy and changes in healthcare providers [34]. Several countries already have e-prescribing infrastructures in place, which could be tailored to meet the data specifications, and allow healthcare providers to monitor a patient’s persistence and intervene to avoid gaps or lapses in medication refills [35,36]. This information could be shared across the primary and secondary care interface, supporting multidisciplinary and multifaceted interventions, which are known to be more effective [37,38]. At the national and regional level, comparative analysis of patterns of medication persistence would also help ascertain the impact of different healthcare policies and interventions on persistence with treatment and ascertain best practice [39].

Integrating pharmacy claims data with more complete data sources, such as clinical and health outcomes, would provide information on the clinical indication for each medication and reasons for medication changes or cessation [39,40]. This integration could also be used to establish the association between gaps or lapses in medication refills and adverse health outcomes and healthcare resource use. This would establish the validity of the process measure of persistence and enable the economic cost to be quantified. A systematic review of the economic impact of medication non-adherence across multiple disease groups, including 14 disease groups, reported an annual adjusted disease-specific economic cost of non-adherence per person from USD 949 to 44,190 (in 2015) [19]. However, the review concluded that differences in methods of adherence measurement made an accurate estimation of the true magnitude of the cost impossible. Standardised measures of *persistence* would enable both the casual association between persistence and adverse clinical outcomes, healthcare resource use, and the economic cost to be quantified. This may enable the introduction of payment systems that reward healthcare providers for improvements in persistence and patient outcomes. [8].

Advances in ICT are also needed when it comes to measuring *persistence* to concurrent multiple medications, as there is no standard approach agreed yet [41]. In recent years, the prevalence of multimorbidity, defined as the co-occurrence of multiple chronic diseases or conditions in a single individual, has increased rapidly affecting more than 60% of people aged ≥65 years [41,42,43,44]. Patients with multimorbidity often require multiple drugs from different classes (i.e., polypharmacy). This fact is associated with a higher risk of inappropriate drug use, underuse of effective treatments, adverse drug reactions, drug–drug interactions and drug duplicates among others [45]. In clinical practice, switching between classes of medications or prescribing multiple medications to treat a single disease are common situations in patients with multimorbidity and they should be considered when assessing *persistence*. For this purpose, some recent methods include an index based on the presence or absence of multiple medications on each day in the observation period, considering medication switching, duplication and overlapping medications [46].

The challenge of accurately measuring *persistence* to multiple medications in patients with multimorbidity is compounded by not knowing if the medications prescribed are actually consumed, given the complexity of dosing and timing schedules of each medication. Moreover, it could happen that the patient does not fill in a prescription but is actually taking the drug, from previous supplies/stock piling. In order to measure *persistence* in a more accurate manner, it may be necessary to include a measure of patient self-reported medication behaviour. Such information could be gathered directly from patients using devices, mobile apps and assisting tools. This would also enable real-time monitoring and feedback to both healthcare professionals and patients trying to detect where difficulties are most significant and when intervention involving more precise measurement is needed; as a consequence, this aids in alerting healthcare professionals about patients who may require some personalised support and making patients more aware and engaged in their treatment [47,48]. Recent reviews of the literature on deprescribing, recommend involving patients in decision making and treatment planning to empower them to be part of the process [49]. Thus, eHealth offers opportunities to transform every step of the patient’s medicine management journey that is critical to improving long-term patient health outcomes [50].

## 5. Conclusions

Although poor medication adherence has been a healthcare issue for several decades, very few countries measure and report on rates of adherence and persistence at the health system level. In this paper, we have proposed measuring *persistence* as an informative and pragmatic measure of medication-taking behaviour, with a view to developing quality and performance indicators of persistence. Monitoring and reporting persistence as a performance indicator of quality of care could help in improving health system efficiency. Persistence measures need to be comparable to benchmarks and assess best practice between countries and interventions. Knowing differences between countries or regions is critical so that lessons can be learned from those countries/regions, including how they have used different policies and interventions. Many challenges remain and it is important that the indicators are clearly defined, measurable and valid and that they adequately reflect the quality of care in disease management. This requires investing in ICT [51,52,53,54,55] solutions that enable healthcare providers to review complete information on patients’ medication-taking patterns, their characteristics and clinical and health outcomes. This would provide healthcare providers with the means to monitor and report on levels of *persistence* and develop patient-centred multidisciplinary interventions to support and engage with their patients. The use of indicators of persistence may also provide transparency at the health system level and the necessary impetus to develop systematic healthcare policy solutions to improve medication persistence, health outcomes and health system efficiency for everyone.

## Figures and Tables

**Figure 1 ijerph-18-04872-f001:**
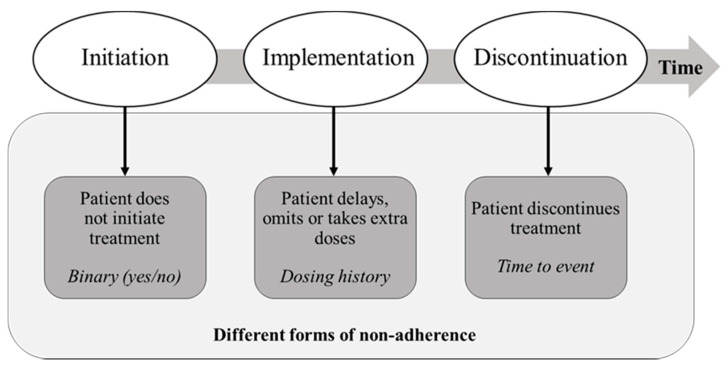
Phases of medication adherence according to the ABC Taxonomy. Based on Vrijens et al., Br. J. Clin. Pharmacol. 2012 [13].

## Data Availability

Not applicable.

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
