# Peer review of "Persistence as a Robust Indicator of Medication Adherence-Related Quality and Performance"

_ijerph, 2021, doi:10.3390/ijerph18094872_

Round 1

Reviewer 1 Report

Congratulations to the authors for the reliable preparation of the proposed measures of compliance with medical recommendations. The proposed indicators are indeed adapted to the prevailing health care systems and can undoubtedly improve the effectiveness of therapy and reduce its costs.

strengths of the manuscript:

well-described indicators for assessing patient compliance.

weaknesses:

I propose to enrich the manuscript with more examples of studies assessing the effectiveness of the above-mentioned assessment indicators. How does the measurement of compliance with medical recommendations in everyday medical practice contribute to the improvement of the effectiveness of therapy, the patient's co-responsibility for the treatment process and reduction of treatment costs for a given disease entity?

Author Response

Response to Reviewer 1

Point 1: Congratulations to the authors for the reliable preparation of the proposed measures of compliance with medical recommendations. The proposed indicators are indeed adapted to the prevailing health care systems and can undoubtedly improve the effectiveness of therapy and reduce its costs.

Response 1: Dear Reviewer, on behalf of all the authors, we sincerely appreciate for stressing the main goal of our work.

Point 2: strengths of the manuscript: well-described indicators for assessing patient compliance.

Response 2: Thanks for your appreciation.

Point 3: weaknesses: I propose to enrich the manuscript with more examples of studies assessing the effectiveness of the above-mentioned assessment indicators. How does the measurement of compliance with medical recommendations in everyday medical practice contribute to the improvement of the effectiveness of therapy, the patient's co-responsibility for the treatment process and reduction of treatment costs for a given disease entity?

 Response 3: Thank you for your suggestion. We have taken steps to further investigate these issues by reporting results from other studies and discussing the importance of linking measures of persistence with clinical and health outcomes, as well as establishing the cost across all disease groups. Please see Lines 225-245 Page 5-6.

Reviewer 2 Report

Dear Authors,

Thank you for submitting this article. I found it to be well written, clear, and specific. It was positive to see the authors identify the key themes in this area, unpack these themes appropriately,  and make some valuable contributions. As an academic pharmacist I found this to be an interesting and useful read. In order to improve this article, I make the following suggestions.

  1. I think the title detracts from the quality of the article, and how interesting it is. I suggest a shortened title with a shift of focus e.g.

    'Persistence as a robust indicator of medication adherence-related quality and performance'

  2. When you mention 'Persistence', it is not initially entirely clear whether using persistence as a performance indicator already exists in the literature or if this is a novel contribution of the paper. This needs to be made explicit as it is an important aspect of the article. I strongly suggest you consider this.
  3. When discussing the limitations and considering medicines adherence is an international problem, it might be interested to hear how 'persistence' as a performance indicator might play out in different international countries.

Author Response

Response to Reviewer 2

Point 1: Dear Authors, thank you for submitting this article. I found it to be well written, clear, and specific. It was positive to see the authors identify the key themes in this area, unpack these themes appropriately, and make some valuable contributions. As an academic pharmacist I found this to be an interesting and useful read. In order to improve this article, I make the following suggestions.

Response 1: Dear Reviewer, we really appreciate your interest in our work.

Point 2: I think the title detracts from the quality of the article, and how interesting it is. I suggest a shortened title with a shift of focus e.g., 'Persistence as a robust indicator of medication adherence-related quality and performance'

Response 2: Thanks for the suggestion. We simplified the title as you suggested.

Point 3: When you mention 'Persistence', it is not initially entirely clear whether using persistence as a performance indicator already exists in the literature or if this is a novel contribution of the paper. This needs to be made explicit as it is an important aspect of the article. I strongly suggest you consider this.

Response 3: We agree with your observation. This paper arises as a result of the publication of the 2018 OECD Report ‘Investing in medication adherence improves health outcomes and health system efficiency’ (Ref 8) aimed to identify factors that are needed for improving adherence to medication at the system level. Hence, the OECD stated that developing and reporting adherence measures as quality improvement indicators or as performance measures could help in improving health system efficiency. Reason why, with the following perspective paper we analyse and propose persistence as an indicator of medication adherence-related quality and performance. We added a clarification of this aspect. Please, see Lines 81-83, Page 2.

Point 4: When discussing the limitations and considering medicines adherence is an international problem, it might be interested to hear how 'persistence' as a performance indicator might play out in different international countries.

Response 4: We implemented this concept in Lines 300-304, Page 8.

Reviewer 3 Report

Overall, very well written and thoughtful discussion on adherence quality indicators. Thank you for the opportunity to review. 

A few thoughts on some areas:

  • Either in introduction or real world challenges - it's worth discussing the financial implication of instituting something like you are proposing of systems communicating effectively. Who would be responsible for paying for this? An example - in the US alone there are hundreds of unique and different opperating systems for hospitals, pharmacies, clinics, etc. To get these all to communicate and share information is something all clinicians would appreciate but this has been estimated be a multi-million (if not billion) dollar process which no company (or government) wants to pay for. This is a large limitation that should be mentioned as it's not as easy as it's portrayed.  
  • I appreciate the note in the real world application that patients actually taking the medications vs. picking them up, is a substantial issue. I may reference this earlier (both section 3 and 4 mention indicators that could be impacted by patients not taking medications that they have received ... a great example is mail order pharmacies in the US that send prescriptions regardless of if the patient requests or not). 
  • Consider adding other measures of non-adherence at the bottom of point 4 to include things like hospitalizations, changes in insurance or providers and that these may impact adherence (similar to how you noted that patient actually taking the medications impacts these results).  

  • The one area that I do think needs some additional discussion or debate is on page 5, line 199 about risk adjustment measures utilized to adjust for things such as low-socioeconomic status. I actually find this to be a poor application of this concept in order for "data" to look nicer. If we "adjust" for low-income status to make our persistence look better, than health care entities would be less likely to address this systemic issue (such as the impact on race on patient care in many countries) by "adjusting" it away. While from a research perspective, this may be a valid method to make data look better, from an ethics perspective this has the potential to widen that gap by adjusting it away instead of helping to identify that gap to show providers and health care entities areas where this financial gap is causing lower persistence to hopefully point out areas where they need to work harder with their patients to help provide access to care.  I understand why you added this, as again it’s a marker that will impact persistence, but I feel that "adjusting" it away will actually drive worse care for the patients who need it most.  I would strongly encourage either dropping this idea or discussing the ethical implications of this suggestion to ensure your manuscript is complete. 

Author Response

Response to Reviewer 3

Point 1: Overall, very well written and thoughtful discussion on adherence quality indicators. Thank you for the opportunity to review.

Response 1: Dear Reviewer, thanks for your positive consideration.

Point 2: A few thoughts on some areas: Either in introduction or real world challenges - it's worth discussing the financial implication of instituting something like you are proposing of systems communicating effectively. Who would be responsible for paying for this? An example - in the US alone there are hundreds of unique and different opperating systems for hospitals, pharmacies, clinics, etc. To get these all to communicate and share information is something all clinicians would appreciate but this has been estimated be a multi-million (if not billion) dollar process which no company (or government) wants to pay for. This is a large limitation that should be mentioned as it's not as easy as it's portrayed. 

Response 2: Thanks for the pertinent comment. Based on the OECD Report published in 2018 (Ref 8), it identifies four enabling factors that are needed to improve medication adherence at the system level: Acknowledge, Inform, Incentivize, and Steer and Support'. The concept of Incentivize encapsulates the very financial issue you discussed, emphasizing how changes in financial incentives for providers and patients are essential. Hence, it stated that, moving to payment systems that reward providers for quality patient outcomes would provide a strong motivation to improve adherence. Medication adherence could also be considered as a measure for performance-based contracts with pharmaceutical companies. The report also recommends that where patient co-payments exist for chronic medications, their reduction or removal should be considered to reduce financial barriers. We added this concept in Lines 225-245 Page 5-6.

Point 3: I appreciate the note in the real world application that patients actually taking the medications vs. picking them up, is a substantial issue. I may reference this earlier (both section 3 and 4 mention indicators that could be impacted by patients not taking medications that they have received ... a great example is mail order pharmacies in the US that send prescriptions regardless of if the patient requests or not).

Response 3: Thanks for suggestion. We provided to also explore this issue in section 4. Please, see Lines 184-188, Pages 4-5.

Point 4: Consider adding other measures of non-adherence at the bottom of point 4 to include things like hospitalizations, changes in insurance or providers and that these may impact adherence (similar to how you noted that patient actually taking the medications impacts these results). 

 Response 4: We added these observations as real world challenges in measuring persistence as these are also important and need to be considered (Lines 212-213, Page 5).

Point 5: The one area that I do think needs some additional discussion or debate is on page 5, line 199 about risk adjustment measures utilized to adjust for things such as low-socioeconomic status. I actually find this to be a poor application of this concept in order for "data" to look nicer. If we "adjust" for low-income status to make our persistence look better, than health care entities would be less likely to address this systemic issue (such as the impact on race on patient care in many countries) by "adjusting" it away. While from a research perspective, this may be a valid method to make data look better, from an ethics perspective this has the potential to widen that gap by adjusting it away instead of helping to identify that gap to show providers and health care entities areas where this financial gap is causing lower persistence to hopefully point out areas where they need to work harder with their patients to help provide access to care.  I understand why you added this, as again it’s a marker that will impact persistence, but I feel that "adjusting" it away will actually drive worse care for the patients who need it most.  I would strongly encourage either dropping this idea or discussing the ethical implications of this suggestion to ensure your manuscript is complete.

Response 5: We appreciate your thoughts and your explanations against the ‘adjustment’ of the socioeconomic status. In order to avoid a misunderstanding of our main intention, such as to discuss the possibility of developing adherence-related quality and performance indicators, we deleted this concept.
